METHODS

# Rhometa: Population recombination rate estimation from metagenomic read datasets

**Sidaswar Krishnan**[1], **Matthew Z. DeMaere**[2]*, **Dominik Beck**[3], **Martin Ostrowski**[1], **Justin R. Seymour**[1], **Aaron E. Darling**[2,4]

**1** Climate Change Cluster, Faculty of Science, University of Technology Sydney, Sydney, NSW, Australia, **2** Australian Institute for Microbiology & Infection, University of Technology Sydney, Sydney, NSW, Australia, **3** Centre for Health Technologies and the School of Biomedical Engineering, University of Technology Sydney, Sydney, NSW, Australia, **4** Illumina Australia Pty Ltd, Ultimo, NSW, Australia

* Matthew.DeMaere@uts.edu.au

**Data Availability Statement:** All supporting information can be accessed here: https://doi.org/10.5281/zenodo.7634208. The Rhometa software package is available at https://github.com/sid-krish/

## Abstract

Prokaryotic evolution is influenced by the exchange of genetic information between species through a process referred to as recombination. The rate of recombination is a useful measure for the adaptive capacity of a prokaryotic population. We introduce Rhometa (https://github.com/sid-krish/Rhometa), a new software package to determine recombination rates from shotgun sequencing reads of metagenomes. It extends the composite likelihood approach for population recombination rate estimation and enables the analysis of modern short-read datasets. We evaluated Rhometa over a broad range of sequencing depths and complexities, using simulated and real experimental short-read data aligned to external reference genomes. Rhometa offers a comprehensive solution for determining population recombination rates from contemporary metagenomic read datasets. Rhometa extends the capabilities of conventional sequence-based composite likelihood population recombination rate estimators to include modern aligned metagenomic read datasets with diverse sequencing depths, thereby enabling the effective application of these techniques and their high accuracy rates to the field of metagenomics. Using simulated datasets, we show that our method performs well, with its accuracy improving with increasing numbers of genomes. Rhometa was validated on a real *S. pneumoniae* transformation experiment, where we show that it obtains plausible estimates of the rate of recombination. Finally, the program was also run on ocean surface water metagenomic datasets, through which we demonstrate that the program works on uncultured metagenomic datasets.

## Author summary

Microbes, specifically prokaryotes, are able to exchange DNA between them through a process called recombination that takes the form of gene-conversion. Recombination plays a fundamentally important role in microbial speciation and evolution. Metagenomics allows us to study microbes in their natural environment as they are via direct sequencing and analysis of environmental DNA. Indeed, most microbes cannot be cultured and can only be studied in this manner. Metagenomic datasets represent an

Rhometa. The metagenomic dataset simulation pipeline is available at https://github.com/sid-krish/rhometa_sim, the LDhat Nextflow Pipeline is available at: https://github.com/sid-krish/Nextflow_LDhat, the full genome version of Rhometa, developed for testing purposes, is available at https://github.com/sid-krish/Rhometa_Full_Genome and the Nextflow_LDhat_sim simulation pipeline (used for simulating full sequences for both Rhometa Full Genome and LDhat Nextflow Pipeline) is available at: https://github.com/sid-krish/Nextflow_LDhat_Sim.

**Funding:** This work was supported by an Australian Government Research Training Program Scholarship. This research was supported by the Australian Government through the Australian Research Council Discovery Projects funding scheme under the project DP180101506, http://purl.org/au-research/grants/arc/DP180101506 (to AED). The funders had no role in study design, data collection and analysis, decision to publish, or preparation of the manuscript.

**Competing interests:** I have read the journal's policy and the authors of this manuscript have the following competing interests: A. E. Darling holds equity in Illumina Inc and is employed by its subsidiary Illumina Australia Pty Ltd, a company that develops and sells DNA sequencing technology. All other authors declare no competing financial interests.

excellent resource for measuring prokaryotic recombination. Rhometa is a new software program that we have designed to be used to interrogate modern metagenomic shotgun sequencing read datasets to estimate the population recombination rates. It extends the composite likelihood approach for population recombination rate estimation, and makes it applicable to modern aligned metagenomic datasets of various depths. The input for the program requires little pre-processing and only an aligned BAM and reference FASTA in the form of a complete sequence or MAG are needed. The program performs well on large BAM files and the included subsampling functionality ensures that files of arbitrarily large size are subsampled and analysed. The program has been validated on simulated datasets and further on experimental and environmental datasets where the amount of recombination is quantified. Through the validation we show that the program is able to reliably estimate the recombination rates in simulated and experimental datasets.

## Introduction

A primary question in the field of microbial ecology is to understand the rate at which prokaryotes evolve and form species in nature. A major driving factor of prokaryotic evolution is recombination [1]. Within prokaryotes, recombination often takes the form of gene-conversion where homologous sequences of DNA are non-reciprocally transferred and replaced with another [2,3]. This process can occur between repeated sequences within the same chromosome and between homologous chromosomal sequences [3]. Recombination often plays a greater role than *de novo* mutation for evolution in prokaryotes [4]. Furthermore, it is thought that recombination plays an important role in counteracting the effects of Muller's ratchet, the theorised process where deleterious mutations inevitably accumulate over time leading to the irrevocable loss of most mutation free genotypes in a population [5,6]. Therefore, understanding the rate at which recombination occurs within prokaryotic populations can provide us insight into a crucial biological process that is necessary for their adaptation and survival.

Currently the most viable way to study microbial populations is via metagenomics, which allows us to study microbes in their natural environment via direct sequencing and analysis of environmental DNA [7,8]. Shotgun metagenomic sequencing yields fragments of DNA sequences, referred to as reads, which taken together represent a random sampling of genome fragments from all the microbes in the environmental sample [9]. These reads can then be used to estimate the rates of recombination.

The rate of recombination within a population can be inferred using population genetic models for evolution. The Wright-Fisher model provides an analytical framework that quantifies various forces that can impact the evolution of a population such as random genetic drift and mutation [10]. Coalescent theory, building on the Wright-Fisher population model, provides an analytical framework for DNA polymorphism data and can be used to obtain quantitative estimates for recombination and mutation rates [11,12].

Coalescent theory provides the population scaled recombination rate for the gene-conversion model of recombination. It is formulated as $\rho = 2N_e r$, or *2 × "effective population size" × "per individual" "per generation" "recombination rate"*, respectively [12] as well as the haploid population scaled mutation rate equation $\theta = 2N_e u$, or *2 × "effective population size" × "per individual" "per generation" "mutation rate"*, respectively [12]. It is difficult to estimate *r* or *u* directly without additional prior information, so recombination and mutation rates are typically computed as the population scaled statistics $\rho$ and $\theta$ or simultaneously as the ratio *r/u* also denoted as r/m (per site recombination to per site mutation rate) [12,13]. An important

point of note is that $r$ and $u$ are per site rates. ρ applies to the entire genome, $\theta$ on the other hand is also per site.

Several approaches have been used to estimate the recombination rate ρ. These include moment estimators, full-likelihood estimators and composite likelihood estimators. Moment estimators use summary statistics to estimate ρ, but their accuracy is limited by the fact that they cannot use all the genetic information available [14–16]. Full likelihood estimators are able to utilise all the genetic information available to them, but are so computationally intensive that their usage is impractical. To mitigate these issues and to make the approach more computationally tractable, composite likelihood estimators were developed [12,16,17]. With composite likelihood estimators, the scope of data that is analysed is reduced e.g. to only consider pairs of alleles, this approach is less computationally intensive with only a slight loss in accuracy compared to the full-likelihood approach [12,16–18].

There are several programs available that implement the composite likelihood approach for estimating the recombination rate, including LDhat [12,19], LDhelmet [20], LDhot [21], PIIM [22] and pyrho [23]. Each are excellent for their respective use cases, but have limitations that make them unsuitable for modern read-based metagenomic datasets.

More specifically, LDhat [12,19], LDhelmet [20] and LDhot [21] were designed for genome sequence analysis, not metagenomes. PIIM [22] was a pioneering attempt at a metagenomic read-based population recombination rate estimator which deals with the complexities of such datasets such as varying depths across loci. While innovative at the time its application is impractical today. PIIM's approach included computationally expensive techniques to integrate out uncertainty in low quality base-calls so as to retain as much information as possible from the scarce data available at the time. Today, deep sequencing is affordable and highly accurate, such that it's often more practical to simply discard low quality sequence data rather than account for it computationally using complex algorithms. As such PIIM's approach is impractical for the ever-larger datasets that are generated via modern sequencing techniques. Furthermore, it lacks support for modern sequence data formats (e.g., BAM), being limited to the obsolete ACE assembly format that is rarely used today.

pyrho [23] is a recent composite likelihood estimator supporting read based data, however it is limited to the analysis of diploid genomes, making it unsuitable for the analysis of haploid genomes (prokaryotes) which commonly dominate metagenomic datasets. Still other programs exist that calculate the population recombination rate through different approaches such as LDjump [18] and CodABC [24] which utilise summary statistics [18], and programs such as ClonalFrameML [25,26] which can provide an estimate of recombination rate relative to the mutation rate, but is designed around bacterial isolate genomes.

mcorr [27] is a program that can work with metagenomic reads and estimate the relative rate of recombination to mutation as well as the recombinational divergence. mcorr uses an alternative mathematical formulation to parameterise the recombination process, with $\phi_{pool} \equiv 2\underline{T}\gamma$, where $\underline{T}$ "is the mean pairwise coalescence time across all loci in the bulk pool" and $\gamma$ the per base pair (bp) per generation recombination rate, equivalent to $r$ in $\rho = 2N_e r$. Furthermore, the program is limited to coding regions and requires a gene annotation file. It is our aim to build on methods established in previous composite likelihood estimators for population recombination rate estimation to create a tailored solution that is applicable to modern aligned read-based metagenomic datasets.

Here, we present Rhometa, a software implementation of the composite likelihood based population recombination rate ($\rho = 2N_e r$) estimation method, which builds upon the methods introduced in the LDhat pairwise program [12] and can be applied directly to modern aligned shotgun metagenomic read datasets for prokaryotes. Details of its implementation are

presented in the methods, while an evaluation of its accuracy on simulated and real data and comparison to existing tools are presented in the results.

## Description of the method

Our approach focuses on advancing the composite likelihood recombination rate estimator for use with aligned metagenomic read datasets. We have built our metagenomic population recombination rate estimator program upon the approach introduced in the LDhat program, specifically the LDhat pairwise module [12]. LDhat is a well-known and used program with support for microbial datasets, specifically for the gene-conversion type recombination which occurs in microbes, however, it is limited to aligned genome sequences. We have designed the program to work with aligned read based metagenomic datasets, where the complication of varying depths and short reads needs to be addressed. For our implementation, we have also subsumed features from pyrho [23]. pyrho, while lacking support for microbial (haploid) datasets, is a modern composite likelihood estimator implemented in python. Like pyrho, our program is also implemented in python and aims to make use of modern libraries and their features. As a result of this shared implementation approach, we were able to call applicable functionalities from pyrho, helping avoid unnecessary code rewrites.

### User Input

We have endeavoured to make the process of preparing a metagenomic dataset for analysis with Rhometa straight forward with few pre-processing steps. Short reads can be aligned to existing reference genomes representing the metagenomic dataset, to reference MAGs (Metagenome-Assembled Genome) or pangenomes representative of the microbial community. Multi sequence references are also supported. The reference genomes provide a scaffold to align the reads, from which the rates of recombination and mutation are determined. For input files, the Rhometa pipeline requires a FASTA format reference sequence and a BAM file of metagenomic reads of interest aligned to the reference. For our pipeline evaluation, we have used BWA MEM (default parameters) to produce the input BAM file [28] where necessary.

### Variant site pairs

The first step of the pipeline involves identifying variant sites (also known as segregating sites). Our program first filters the user supplied BAM for mapping quality and relative alignment score and subsequently performs variant calling against the user supplied reference FASTA using the program freebayes (default parameters with -p (ploidy) = 1) [29]. The resulting VCF file, containing information on all predicted variant sites, is reduced to only single nucleotide polymorphisms (SNPs) using bcftools [30].

Rather than individual variant sites, the composite likelihood estimator as implemented in LDhat considers variant site pairs, tracking count and position within the reference genome's coordinate space to estimate the recombination rate. For instance, if variant sites are found at reference positions 1, 3, and 5 the set of variant site pairs would then be (1, 3), (1, 5), and (3, 5).

### Pairwise table

The LDhat pairwise module was designed for genome sequences and considers all possible variant site pair combinations across the sequences being analysed. Rhometa restricts its consideration to the set of variant site pairs linked by individual reads or read-pairs. For single-end reads, both sites within a variant pair must fall within the extent of an individual read, while for paired-end reads variants can fall within the insert length. A separation limit of 1000

**Table 1. Pairwise table example.**

|  | AA | AC | AG | AT | CA | CC | CG | CT | GA | GC | GG | GT | TA | TC | TG | TT |
|---|---|---|---|---|---|---|---|---|---|---|---|---|---|---|---|---|
| (130, 136) | 0 | 13 | 0 | 21 | 0 | 0 | 0 | 10 | 0 | 0 | 0 | 0 | 0 | 0 | 0 | 0 |
| (130, 143) | 13 | 0 | 21 | 0 | 0 | 0 | 10 | 0 | 0 | 0 | 0 | 0 | 0 | 0 | 0 | 0 |
| (130, 169) | 0 | 29 | 1 | 3 | 0 | 8 | 0 | 0 | 0 | 0 | 0 | 0 | 0 | 0 | 0 | 0 |
| (130, 311) | 2 | 0 | 26 | 0 | 0 | 0 | 9 | 0 | 0 | 0 | 0 | 0 | 0 | 0 | 0 | 0 |
| (130, 358) | 0 | 0 | 0 | 19 | 0 | 2 | 0 | 6 | 0 | 0 | 0 | 0 | 0 | 0 | 0 | 0 |

bp is imposed on paired end variant site pairs reflecting a practical upper limit on insert size for current Illumina short-read sequencing technology [31]. Rhometa performs well with both single and pair-end reads, with very little difference in the results between the two (S2 Fig).

For all accepted variant site pairs, we construct a pairwise table of observational frequency (Table 1). The pairwise table allows for the possibility of all 16 combinations for any variant site pair. The table also captures the fact that multiple reads can align at a position. Instances where variant site pairs contain an ambiguous base (eg. N) are ignored.

## Splitting the pairwise table by depth

In the pairwise table, variant site pair total alignment depth is calculated by row summation (e.g., For the pair (130, 136) from (Table 1), total depth is 13 + 21 + 10 = 44. For the whole genome approach of LDhat, this marginal value is a constant, while for metagenomic data depth of coverage can vary greatly across sites. As such it is necessary to split the pairwise table into constant depth subtables so that the depth can be taken into account and handled in downstream processing.

## Bi-allelic pairwise table

For each constant depth subtable, sites that do not contain two alleles are excluded (only biallelic sites should be in the pairwise table).

## Lookup tables

Lookup tables improve the computational efficiency of the composite likelihood approach by precomputing the likelihoods for different configurations of sets of allele pairs. Lookup tables are generated under a fixed population mutation rate and a range of population recombination rates, typically between 0–100 [12,19]. We use the program LDpop [32] for generating lookup tables as it is the most feature rich and most efficient program of its kind currently. Details on how the lookup tables are used can be found in S1 Appendix. It is a standard process for which we have made use of some functions from pyrho to avoid reimplementation. Generation of lookup tables with LDpop are parameterised by the number of genomes, range of population recombination rates, and $\theta$ per site. Further, the "approx" option is used which is significantly faster but still quite accurate when compared to the ldpop's exact algorithm. This also makes generation of large lookup tables, as is necessary with Rhometa, more tractable. Generally, the generation of large lookup tables is a computationally intensive process, more information on which can be found in the paper associated with LDpop [32].

## Watterson's theta estimate

A subprogram is provided to estimate the population mutation rate ($\theta = 2N_e u$), per site. The formulation is directly based on Watterson's theta estimate as implemented in LDhat [12],

with some changes on the information used for the input parameter. The program requires the aligned BAM file and the reference FASTA file and makes use of freebayes to identify variant sites which is required for the Watterson estimate. $\theta$ is a required parameter for lookup table generation adjusted for read based datasets, the $\theta$ estimate is calculated based on dataset depth–specifically mean and median depth–in place of the number of genomes, as in the original formation (S4 Fig). For metagenomic samples where the exact number of genomes is unknown, the true number of different genomes in a metagenomic sample is likely to be a large value, as such using the depth value is reasonable and in many cases is likely to be a conservative estimate for the number of genomes.

## Lookup table and depth

The size of each constant depth pairwise sub-table determines the lookup table to match against for precomputed negative log-likelihoods. In the context of Rhometa and aligned reads, the number of genomes parameter used to generate lookup tables is instead taken as depth. Where the bam file is of a high depth, this can result in pairwise sub-tables of high depth. In such cases, available lookup tables may not be of high enough depth to cover them. A subsampling feature is included that is able to automatically downsample the BAM to a given depth. This ensures that positions with a depth exceeding that of the highest generated lookup table are still evaluated and are not omitted from consideration. BAM subsampling uses a random sampling process and permits a list of seed values for testing and identifying any variance that can stem from the subsampling. In general, if the depth of the largest available lookup table is small, an increased need for downsampling could result in a decrease in estimation accuracy.

## Calculating $r_{ij}$

The next step is to calculate recombination rate values for each variant site pairs, these values are denoted by $r_{ij}$, with $i$ and $j$ referring to the variant sites. The method used for calculation differs for crossing-over and gene-conversion modes of recombination. prokaryotes undergo recombination via gene-conversion and the equation used to calculate $r_{ij}$ is as follows [12]:

$$r_{ij} = 2ct\left(1 - e^{-\frac{d_{ij}}{t}}\right) \tag{1}$$

For Eq 1, $c$ represents the per base recombination rate, $t$ the mean gene conversion tract length and $d_{ij}$ the distance between a variant site pair. $ct$ is taken together and represents the range of population recombination rates being evaluated, this is typically between 0–100 and is the same as the range of $\rho$ values used when generating the lookup tables. The process essentially involves computing $r_{ij}$ for each variant site pair for the range of population recombination rate values.

## Final pairwise likelihoods

Next, we bring together the information we have generated thus far: the output of the lookup table and depth step and calculating $r_{ij}$ step. For each variant site pair, we use the corresponding negative log-likelihood values from the lookup table and depth step, and on these apply linear interpolation to determine the negative log-likelihood values for $r_{ij}$ value for that variant pair configuration. The interpolation is performed against the range of population recombination rates used to generate the lookup tables. This process is done for all the variant site pairs and the results are the final negative log-likelihoods for a given range of population recombination rates being evaluated, which again is typically between 0–100.

## Population recombination rate

Prior to computing the final log-likelihood sums we weight adjust the negative log-likelihoods. As the number of observations provided at a given depth represents the degree of evidential support towards the final ρ estimate, we introduce a novel weighting algorithm that accounts for the additional information in high depth, high observation count site pairs relative to low depth / low count site pairs. The weighting algorithm is a simple solution to reliably add more weight to higher depth regions and the number of observations at that depth (there can be many regions of same depth). The weighting and final negative log-likelihood summation algorithms are as follows:

$$w_d(\rho) = \ln\left(\frac{\exp T_d}{\sum_\rho \exp T_d} dn_d\right) \tag{2}$$

$$\rho_{max} = argmax\sum_d w_d(\rho) \tag{3}$$

Here, $d$ represents depths observed in the dataset. Weighting is performed on each per-depth table denoted by $w_d(\rho)$ (Eq 2). In the right side of the Eq 2, $T_d$ is the unweighted per-depth table and $n_d$ the number of unique variant site pairs. The reweighted negative log-likelihoods $w_d(\rho)$ are collected across depths and summed with respect to the range of population recombination rates being evaluated. The maximum negative log-likelihood value (closest to 0) then corresponds to the final population recombination rate $\rho_{max}$ (Eq 3) estimate.

## Program structure

The program is organised into 4 pipelines, each dedicated to a specific task. These pipelines are written using nextflow, a framework for pipeline management [33]. All the scripts used in the individual pipeline steps were written using the python programming language and various python libraries. Some python scripts were adapted or used as is from the programs LDpop [32] and pyrho [23]. Additional programs used in the pipelines include msprime [34], ART [35], BWA MEM [28], seaborn [36] and samtools [30].

The four pipelines, sim_gen, theta_est, lookup_table_gen and rho_est (Fig 1), correspond to the nextflow pipeline names, e.g. sim_gen.nf within Rhometa, and perform the functions defined in the following paragraphs.

Sim_gen (Fig 1A) is used to generate BAM files and FASTA reference files with simulated reads from bacterial genomes with recombination. The bacterial genomes are simulated using msprime. This pipeline is primarily included so that the simulated datasets used for this paper can be reproduced, but is not required to analyse real datasets. It is in a separate repository and can be accessed at: https://github.com/sid-krish/Rhometa_sim

Theta_est (Fig 1B) is used to determine the population mutation rate per site (θ) based on the Watterson estimate as implemented in LDhat, details in methods. This pipeline estimates θ on the dataset of interest, furthermore, θ per site is one of the required parameters for generating lookup tables. The user has the option to use the estimated θ or a different value when generating lookup tables.

The Lookup_table_gen (Fig 1C) component of the pipeline makes use of LDpop and pyrho to generate the lookup tables required for the recombination rate estimator and can be launched in one of 2 ways. It can either use a pre-generated lookup table for high depth, which then will be downsampled for each depth from 3 to the depth of the lookup table or the pipeline can generate a high depth lookup table from scratch and then perform the downsampling step. The downsampling algorithm is a part of pyrho, it is significantly faster to generate the

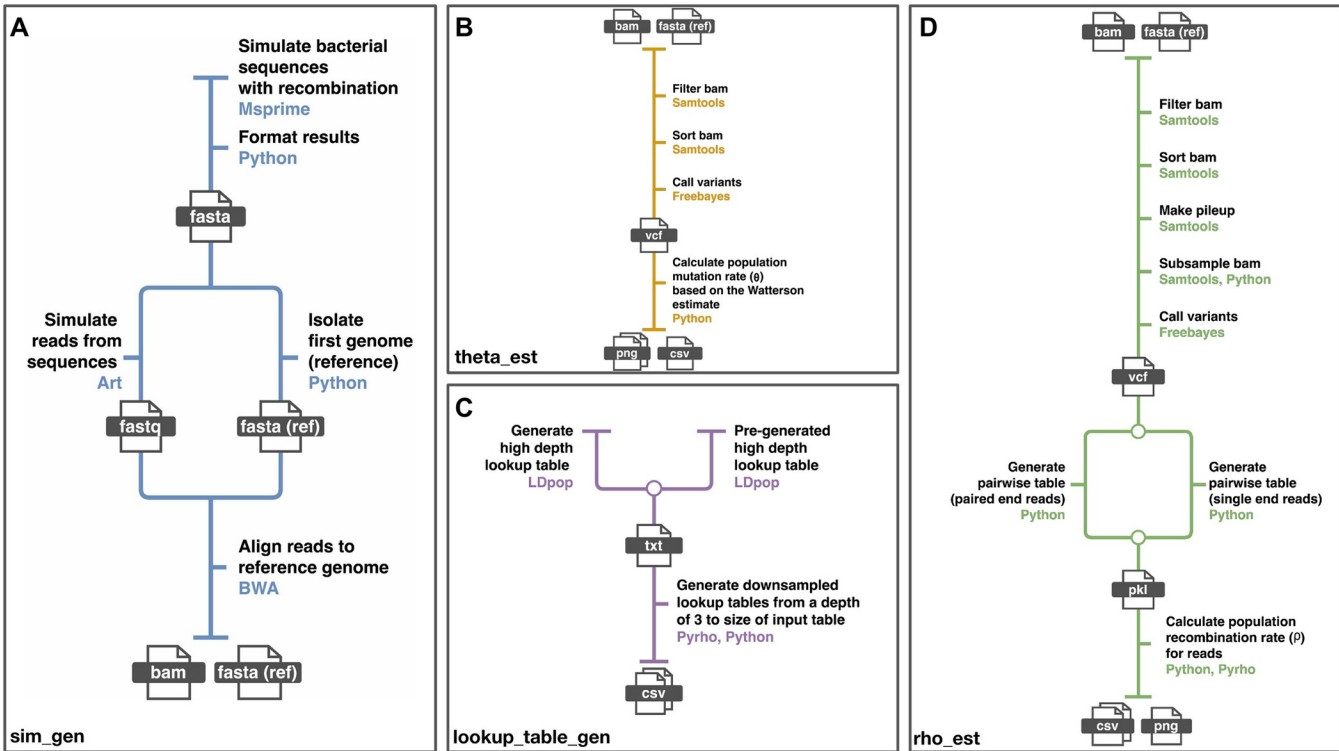

**Fig 1. The pipelines that together make up Rhometa.** (**A**) Pipeline for generating simulated metagenomic read datasets. (**B**) Pipeline for estimating the population mutation rate (**C**) Pipeline for generating the lookup tables required for the recombination rate estimator (**D**) Pipeline for estimating the population recombination rate.

required smaller lookup tables from a larger table via downsampling and the results are essentially identical.

The rho_est pipeline (Fig 1D) is used to estimate the population recombination rate of metagenomic read based datasets provided in the form of BAM and reference FASTA files. It makes use of the lookup tables generated by the lookup_table_gen pipeline.

Rhometa is available at: https://github.com/sid-krish/Rhometa

Our pipelines for evaluating LDHat, the Rhometa_full_genome pipeline and the simulated dataset generator for these pipelines can be accessed here:

- LDhat Nextflow Pipeline: https://github.com/sid-krish/Nextflow_LDhat

- Rhometa Full Genome Pipeline: https://github.com/sid-krish/Rhometa_Full_Genome

- Nextflow_LDhat_sim (used for both Rhometa Full Genome and LDhat Nextflow Pipeline): https://github.com/sid-krish/Nextflow_LDhat_Sim

## Simulated datasets

The development of our program was performed in two major phases, for the first phase we endeavoured to create a full genome recombination rate estimation pipeline for bacterial sequences based on the LDhat methodology (Rhometa_full_genome), once we were certain that we were able to replicate LDhat's results exactly we then carefully adapted the program to work with read based datasets (Rhometa).

To evaluate LDhat and Rhometa_full_genome, we utilised msprime [34] to simulate bacterial sequences with recombination. Our simulations included multiple genomes (5–100

genomes) of size 25KB, under population recombination rates [5, 12.5, 25, 37.5, 50], mean recombination tract length 500bp, with 10 replicates (seed values 1–10) and population mutation rate 0.01. Lookup tables for population mutation rate 0.01 and population recombination rates 0–100 (101 steps) were used.

The LDhat pipeline configured for gene-conversion is available at: https://github.com/sid-krish/Nextflow_LDhat. Rhometa_full_genome pipeline is available at: https://github.com/sid-krish/Rhometa_Full_Genome. The full genome simulation pipeline is available at https://github.com/sid-krish/Nextflow_LDhat_Sim. A point of note is that the $\theta$ estimator is implemented separately by us as per [12, Eq 1] in both our LDhat pipeline and Rhometa_full_genome pipeline. Additionally, all variant sites are used for $\theta$ estimation, not just bi-allelic ones.

When simulating the population recombination rate with msprime, the number of samples (genomes), sequence length, recombination rate ($r$), mean gene conversion tract length ($t$), seed value and mutation rate ($u$) are provided, the effective population size ($N_e$) was 1 (default) and the ploidy ($i$) was fixed to 1. Default options are used in all other cases. Within msprime, the population recombination rate $\rho$ is calculated as such: $2 * i * N_e * r * t$. The per site population mutation rate $\theta$ was calculated as such: $2 * i * N_e * u$.

Initially the number of genomes was fixed and we varied the length of the genomes, but this analysis revealed that varying the genome length does not have a significant impact on the final population recombination rate estimations (S1 Fig). We therefore fixed the genome length and varied the number of genomes and in doing so we found that as the number of genomes increased the accuracy and variance of the final estimations also improved (Fig 2).

We took a similar approach to evaluating the read based pipeline Rhometa to that used with LDhat and Rhometa_full_genome. For the read-based pipeline, the simulated full bacterial sequences, simulated via msprime, are further processed to be in the form of reads using the read simulator ART [35], which simulates sequencing reads by mimicking real sequencing processes with empirical error models. The reads were simulated based on the Illumina HiSeqX TruSeq system. These reads are then aligned to one of the bacterial sequences which represents the reference FASTA file, the first of the simulated sequences is used for this (Fig 1A). The aligned BAM and reference FASTA are then used for recombination rate estimation.

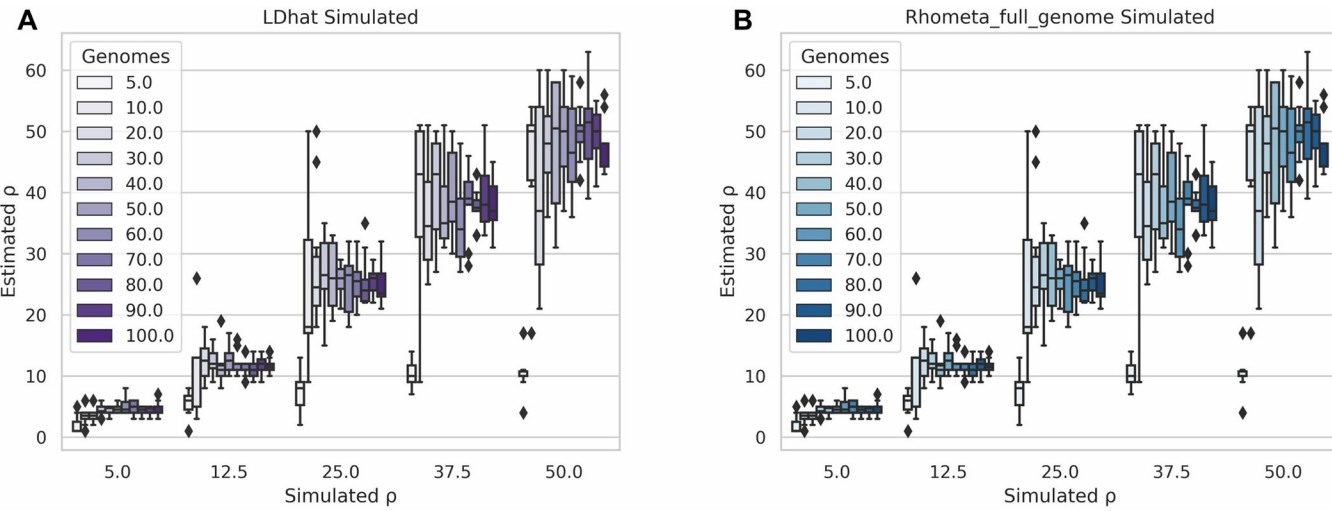

**Fig 2. Comparison of LDhat and Rhometa_full_genome when running on simulated full genomes.** (**A**) LDhat. Simulated vs Estimated population recombination rate ($\rho$) for varying number of simulated full bacterial genome sequences. (**B**) Rhometa_full_genome. Simulated vs Estimated population recombination rate ($\rho$) for varying number of simulated full bacterial genome sequences.

The simulation parameters were as follows: the effective population size was 1 (default) and the ploidy was set to 1, number of genomes 20–200, genome length 100KB, population recombination rates [10.0, 20.0, 30.0, 40.0, 50.0], mean recombination tract length 1000bp, with 20 replicates (seed values 1–20) and population mutation rate of 0.01. Each seed value used applies to all aspects of the pipeline where a seed is required. The reads were paired-end of length 150bp, insert length 300bp, standard deviation of 25bp, with window size set to 1000 during analysis and the fold coverage values were [1, 4, 8, 16]. The lookup tables were generated and used for 3–250 (genomes), generated under population mutation rate 0.01 for population recombination rates 0–100 (0–1 in 101 steps plus 1–100 in 100 steps). Bam subsampling was also automatically applied by Rhometa during analysis if needed.

Additionally for the read-based pipeline, we evaluated the deviation of the estimated results from the simulated values. The formula used to calculate the deviation is (Estimated ρ (mean) —Simulated ρ) / Simulated ρ. This makes it easier to gauge the magnitude of deviation from the expected.

## Real Datasets—Transformation experiment

To further evaluate Rhometa we applied our pipeline on the data derived from a previously published laboratory transformation experiment, where the extent and distribution of recombination events were quantified. In the experiment [37], in vitro recombination through transformation was performed on a S. pneumoniae strain. Transformed isolates were then sequenced and recombination events were identified. This dataset was also used to evaluate the mcorr method by its authors and as such it provides us with the opportunity to compare the results of our pipeline against those published in the mcorr paper.

The transformation experiments were performed with different concentrations of donor DNA, 5 ng mL-1 and 500 ng mL-1, 5 ng mL-1 and 500 ng mL-1 experiments (S2 Table) had a similar number of recombination events, with the 5 ng mL-1 having a slightly larger number of events, the authors state that this indicates a single piece of DNA can act as the origin for multiple recombination events. The dataset is available in the form of reads, which Rhometa was designed to analyse. Each 5 ng mL-1 sample from experiment 1 was aligned to S. pneumoniae reference sequence ATCC 700669, NCBI accession NC_011900.1 the resulting BAM files were then merged and analysed with Rhometa.

To analyse the datasets, we first estimated the $\theta$ for median depth using the $\theta$ estimation pipeline, from which we obtained $\theta$, that is per site by default. We then generated lookup tables, based on the $\theta$, for population recombination rates 0–20 in 201 steps for 3–200 genomes and used the lookup tables for the recombination rate estimation pipeline. Subsampling was enabled, with a window size of 1000 for paired end reads. As given in the Croucher et al paper [37], we used the value of $t$ = 2300 bp as the mean tract length for analysis. Additionally, we used 5 different seed values [0, 1, 2, 3, 4] for the subsampling step to account for any variance and then took the average of the values for recombination rate estimations.

Using the ρ and $\theta$ estimates along with information from the experiment we also calculated the ρ per site and r/m values. The default ρ estimate, by Rhometa, is a whole-genome estimate. To obtain ρ per site, the estimated ρ value was divided by the mean tract length of 2300 bp. To get the r/m value, we used the conversion formula [26]: ρ (per site)/$\theta$ (per site) * tract length * substitution probability. We estimated the substitution probability between the donor and recipient and found it to be $\frac{17534-385}{2221315} = 0.00772$, where 2221315 is the recipient genome length and based on the information provided by Croucher et. al (2012) ". . . the donor DNA identified 17,534 SNPs when aligned to the recipient sequence, of which 385 appeared to be false positives . . . These positions were excluded from subsequent analyses." [37].

We repeated the process above for each 500 ng mL-1 sample from experiment 1 and the final merged BAM was analysed with Rhometa. Furthermore, the 5 ng mL-1 samples and 500 ng mL-1 samples in experiment 1 were analysed together, corresponding to 84 sequences. We prepared this dataset by merging the final 5 ng and 500 ng BAMs. We performed this analysis, to enable comparison with mcorr's published results. The analysis was performed using the same process as with the 5 ng mL-1 samples and 500 ng mL-1 samples.

### Real datasets—Ocean metagenomic dataset

To evaluate our pipeline on typical metagenomic data we selected 15 publicly available short-read metagenomic reads-sets (S3 Table) sampled from waters at the Australian Integrated Marine Observing System (IMOS) Port Hacking National Reference Station (NRS) [38] (S3 Table). These 15 datasets represent 15 consecutive monthly time points (July 2012—September 2013). Next, we assembled a long-read metagenomic read-set (accession: SRR13002033) collected in March 2018 from the location using metaFlye (v2.8.3-b1705) (default options) [39], Three long contigs (lengths: 0.8–1.0 Mbp) were selected from the assembly to be used as references for read-mapping (Table 3). Confident taxonomic assignments were determined for each contig using MMseqs2 (v14.7e284) (default easy-taxonomy options) [40] against the latest GTDB taxonomic database (release: R07-RS207) [41]. The 15 read-sets were mapped to each reference contig using BWA MEM (v0.7.17-r1188) (default options) [28] and the resulting BAMs merged respectively with samtools (v1.16.1) [30]. CoverM (v.0.6.1) [42] was then used with the metabat method to determine the coverage for the aligned bams. Lookup tables for 3–250 (genomes), generated under population mutation rate 0.01 for population recombination rates 0–100 (0–1 in 101 steps plus 1–100 in 100 steps) were used. The tract was fixed at 1000bp as was the window size. As with the transformation experiment analysis we used 5 different seed values [0, 1, 2, 3, 4] for the subsampling step to account for variance and then took the average of the values for ρ estimations.

## Verification and comparison

### Evaluation on simulated datasets

We first validated the full genome version of Rhometa (Rhometa_full_genome), which reimplements the core LDhat pairwise method to estimate ρ. This was done to ensure accuracy in reimplantation of core LDhat algorithms which forms the basis for the read based (Rhometa) implementation. Comparison of estimated population recombination rate (ρ) between LDhat and our reimplementation (Rhometa_full_genome), using our sweep of simulated genomes, shows identical results between LDhat (Fig 2A) and our reimplementation (Fig 2B), thus ensuring that we have captured LDhat's algorithms accurately. With LDhat and our reimplementation, the number of genomes simulated has a large impact on the accuracy of the estimates, with results improving with higher numbers of genomes, especially at higher recombination rates.

We next evaluated Rhometa's performance using our sweep of simulated read-sets. The number of simulated genomes had a large bearing on estimation accuracy (Fig 3), as also observed with LDhat, accuracy improved as the number of genomes increased and inter-replicate variance decreased as the coverage (fold_coverage) improved. This is especially evident for higher recombination rates. Larger population recombination rate values appear to require a relatively higher number of genomes for accurate estimation. For very low recombination rates between 0–1 (S3 Fig), the improvement in accuracy was not seen and a tendency to overestimate was observed.

To get a clearer picture of the deviation of the estimated population recombination rate from the expected result, we calculated the deviation for the read based results (Fig 4). Here values closer to 0 indicated better performance, while values above 0 are overestimations, and values below 0 are underestimations (i.e. a deviation value of +/- 0.1 would indicate that the final result is off by 10%). As the number of simulated genomes increased, the deviation of estimated to expected tended to decrease, achieving a deviation of less than 5–10% in most cases for a simulated ρ of 50 with 200 genomes and 16x coverage. Such improvement is consistent with the patterns observed in LDhat. For simulated population recombination rates between 10–50, having greater than 80 genomes produced the least amount of deviation (generally within 20–30%), with the results significantly improving when more genomes are present. Our pipeline appears to be robust to variance in fold coverage. The differences between 16x coverage and 1x coverage being minor (Fig 3).

## Evaluation on Real Datasets—Transformation Experiment

After establishing the performance of Rhometa on simulated datasets, we interrogated its performance on real short-read sequence data from a lab-based experiment designed to track and study recombination in *S. pneumoniae* (Table 2), seed averaged values were taken for the final results, refer to S1 Table for individual seed results.

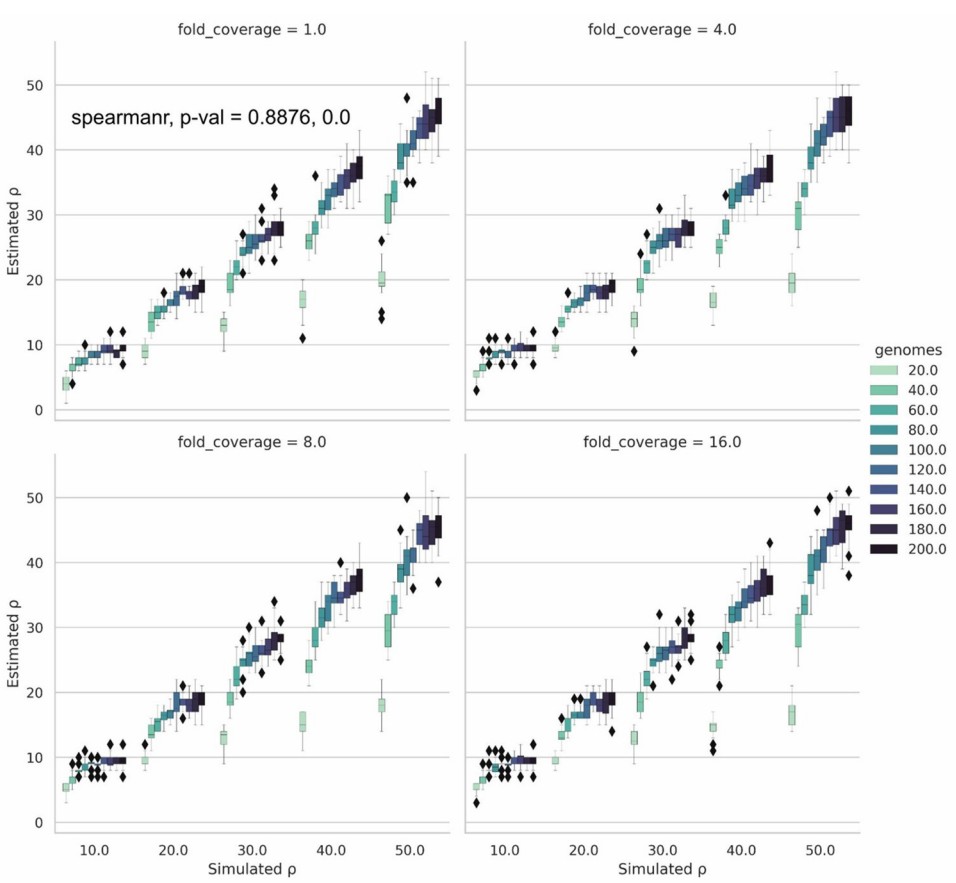

**Fig 3. Simulated vs Estimated population recombination rate (ρ) results for Rhometa.** Results for varying numbers of simulated genomes and fold coverage values for population recombination rates 10.0, 20.0, 30.0, 40.0, 50.0.

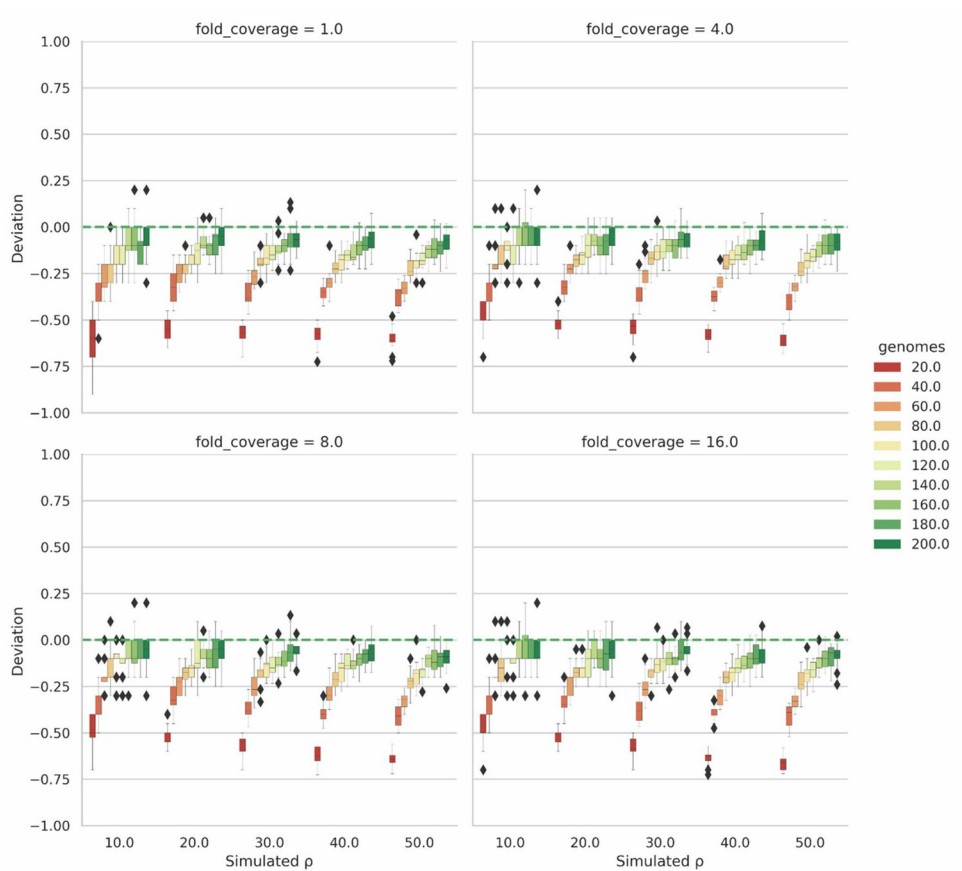

**Fig 4. Deviation plot for results in Fig 3. Deviation is calculated as (Estimated ρ (median)—Simulated ρ) / Simulated ρ.** Deviation results corresponding to Fig 3 for population recombination rates 10.0, 20.0, 30.0, 40.0, 50.0.

To compare the results for the same transformation experiment with those of mcorr, the estimated r/m values were used. The authors of mcorr provide $\gamma/\mu$ (similar to r/m) for the evolved strain (reads) representing combined 5 ng mL$^{-1}$ samples and 500 ng mL$^{-1}$ samples in experiment 1 (84 sequences) where they estimate a $\gamma/\mu$ value of 0.93. Due to the nature of the experiment, it was expected that the rate of recombination would be far higher than the rate of mutation. As an experiment designed to induce transformation over short timescales, this should lead to a large excess of substitutions derived from recombination events, relative to *de novo* mutation. As such, mcorr potentially significantly underestimated the true value and Rhometa better reflects our *a priori* expectation for the 84 sequence dataset with an estimated r/m ratio of 1922.97.

The 84 sequence dataset was also used to benchmark Rhometa. The bam file for this dataset is 11.42 GB in size. On a system with 4 CPUs and 32 GB RAM the rho_est pipeline took 1h 45m to complete and the theta_est pipeline took 2h 45m to complete.

**Table 2. *S. pneumoniae* transformation experiment analysis.**

|  | ρ | ρ (per site) | θ (per site) | ρ/θ (per site) | r/m |
|---|---|---|---|---|---|
| 5 ng mL$^{-1}$ | 5.56 | 0.00242 | 1.8e-5 | 134.4 | 2386.4 |
| 500 ng mL$^{-1}$ | 5.22 | 0.00227 | 2e-5 | 113.5 | 2015.3 |
| 84 sequences | 4.48 | 0.00195 | 1.8e-5 | 108.3 | 1922.97 |

**Table 3. Port Hacking analysis.**

| Contig_ID | edge_21626 | edge_14330 | edge_29441 |
|---|---|---|---|
| Lineage | Cyanobacteria, Cyanobacteria, PCC-6307, Cyanobiaceae (A) | Cyanobacteria, Cyanobacteria, PCC-6307, Cyanobiaceae (B) | Actinobacteriota, Acidimicrobiia, Actinomarinales, Actinomarinaceae |
| Coverage depth | 40 | 25 | 729 |
| Contig length | 1029283 bp | 926756 bp | 797490 bp |
| $\rho$ (Seed 0) | 2.00 | 0.38 | 0.18 |
| $\rho$ (Seed 1) | 0.96 | 1.00 | 0.21 |
| $\rho$ (Seed 2) | 0.94 | 0.41 | 0.22 |
| $\rho$ (Seed 3) | 0.90 | 0.28 | 0.17 |
| $\rho$ (Seed 4) | 1.00 | 0.69 | 0.13 |
| Mean | 1.16 | 0.55 | 0.18 |
| Stddev | 0.47 | 0.29 | 0.04 |

We can calculate the expected average r/m for a genome, as follows. In the Croucher et al paper [37], they state that the mean proportion in the recipient genome changed due to recombination was 1.4% using which we can estimate the average number of bases changed by recombination in a single genome as: 2221315 (recipient genome length) * 0.014 * 0.00772 (substitution probability as calculated in methods above) = 240.08. Also, in the paper it is stated that there were 2,312 polymorphic sites, 59 of which not coming from the donor, 6 of these sites were false positives, with the others likely being de novo point mutations or intragenomic recombinations. So we can take the upper bound for de novo mutations to be 53, then to get the average number of de novo mutations per genome we can divide by 84, which is the total number of sequences in the combined 5 and 500 ng dataset, 53/84 = 0.631.

Using this information derived for the average number of bases changed by recombination and mutation in a single genome, we can calculate the expected average r/m for a genome as 240/0.631 = 380.3, for the combined 84 sequence dataset. The actual value should be higher due to the fact that a region can experience multiple recombination events, so this would be a lower bound estimate. With Rhometa we do observe a value greater than 380.3 of 1922.97 for the 84 sequence dataset.

### Evaluation on Real Datasets—Ocean Metagenomic Dataset

Rhometa was further tested on real datasets, using samples from Port Hacking, Sydney. We produced estimates using reads mapped to 3 different assembly contigs (Table 3). A point of note is that within the family *Cyanobiaceae*, there is a difference in rates of recombination. It is in such comparisons that Rhometa could prove to be of great value.

### Discussion

Recombination plays a crucial role in microbial evolution and speciation [43–45]. Understanding the rate at which recombination occurs provides us an insight into the impact of this process. Metagenomics is the only method that allows us to study recombination in real-world natural microbial communities without culture bias [46]. However, there are currently no software tools to accurately estimate population recombination rates on large metagenomic datasets. To fill this gap, we have developed Rhometa, a software implementation that builds on the composite likelihood estimator for population recombination rate estimation method and enables interrogation of next generation sequencing reads from shotgun metagenomic

experiments. It is adjusted for gene-conversion type recombination as experienced by prokaryotic populations.

Composite likelihood population recombination rate estimators are among the most accurate methods known, and our implementation makes these methods available to the wider metagenomics community. This is significant as most microbes cannot readily be cultured, by some estimates only 1–15% are readily cultivable in laboratories [47].

Shotgun metagenomics yields reads from microbes taken directly from the natural environment and mitigates issues related to culture dependent studies. However, there has, until now, not been a viable approach for quantifying the population recombination rate from these reads. PIIM [22] and mcorr [27] come closest to being applicable to shotgun metagenomic datasets, being designed for this use case. However, in the case of PIIM its statistical model uses a very compute intensive approach to account for low quality data, and missing data, including missing coverage, making the compute requirements impractical for large modern datasets.

Meanwhile, with mcorr, the mathematical formulation is distinct from the well-known population recombination rate ($\rho = 2N_e r$), which may represent challenges for interpretation. On an experimental dataset where transformation was used to produce a population of recombinants for sequencing, the approach implemented in mcorr appears to significantly underestimate the recombination rate. Furthermore, mcorr was evaluated on simulated datasets used to test for Rhometa (S7 Fig) where we expected to see an increase in the $\phi_{\text{pool}}$ estimated as the simulated $\rho$ increases. However, the results showed no correlation between the recombination rate inferred by mcorr and the true simulated rate. Additionally, $\phi_{\text{pool}}$, the value for recombination estimated by mcorr complicates interpretability by not using the standard parameter for population recombination ($\rho$). mcorr's limitation of looking at coding regions provided in a gene annotation file again complicates analysis as this information is not always available. Rhometa builds on established methods and theoretical foundations, without the need for gene annotation files and based on our testing, produces accurate results.

To build our program, our approach was to first reimplement the LDhat pairwise program for gene conversion. Doing so we were able verify that we accurately captured the core algorithms of LDhat while having a modern and adaptable implementation around it. The simulation results for LDhat and the Rhometa_full_genome (Fig 2A and 2B) show that we were able to reproduce the LDhat results 1:1. Having validated the reimplementation we then adapted it for read based datasets. LDhat is effective at detecting changes in the magnitude of the simulated population recombination rates, and produces accurate estimates for cases with large numbers of genomes (Fig 2A). Our analysis showed a trend where the accuracy and variance of the estimates improved as the number of genomes increased.

We then evaluated the performance of Rhometa on simulated datasets and the results (Figs 3 and 4) demonstrated that the read-based pipeline performs well and consequently represents a successful implementation of the composite likelihood population recombination rate estimator for metagenomic read-based datasets. As with LDhat and Rhometa_full_genome, the performance of the read based pipeline improves with the number of genomes present, having 80 genomes or more produces the best results. However, it should be noted regardless of the number of genomes there is a bias towards underestimation of $\rho$, though this is small when there are many genomes present (Fig 4). Very small $\rho$ values, those between 0–1 (S3 Fig), are an exception as the implementation has a tendency towards over estimation.

Rhometa was further applied to a *S.pneumoniae* transformation experiment [37], where the extent of recombination could be directly quantified. This dataset was also analysed by the authors of mcorr for their paper, enabling a cross comparison between the two programs. Preparing and analysing the dataset (Table 2, 84 sequences) in a manner performed by mcorr, it

was found thatRhometa was able to accurately detect the higher rate of recombination relative to mutation as was expected for the dataset. From direct evidence, based on the information provided in the transformation experiment paper, we calculate a conservative lower bound for the ratio of recombination to mutation as $r/m > 380.3$. Rhometa was able to meet this condition by estimating a $r/m$ value of 1922.97, while mcorr estimated a $r/m$ value of 0.93, which suggests that change due to mutation was greater than recombination, which is extremely unlikely.

Evaluation on real environmental metagenomic samples (Table 3) showed that the program works on real datasets. While the analysis performed here was purely to test the applicability of the pipeline, it demonstrates potential for many comprehensive studies. Such as looking at the rate of recombination of a given species over time or across locations.

Rhometa is well positioned to exploit the abundance of preexisting metagenomic datasets to enable a thorough first-pass study of recombination rates in microbial communities.

The main difference between the LDhat approach and the read based approach is as follows. In LDhat the final population recombination rate estimate "is obtained by combining the likelihoods from all pairwise comparisons" [12], the likelihoods (negative log-likelihoods) come from the pregenerated lookup tables as mentioned in methods. For genome sequence datasets this means we can use a likelihood table generated for the exact number of genomes/depth in a dataset, the number of genomes represents the depth which is fixed and all pairs of sites we look at will have this depth. Additionally, LDhat considers all possible variant site pair combinations across the sequences being analysed.

A complication associated with read-based datasets is that the depth can vary greatly from site to site. We addressed this issue by using an appropriate depth lookup table for the variant site pairs being considered, the rationale for which is that the negative log-likelihoods for the variant site pairs considered is obtained individually and then combined for a final composite negative log-likelihood. Taking into account that the negative log-likelihoods are obtained individually for each pair of variant sites, the variant pair combinations considered are limited to the extent of a read for single-end reads, for the insert length in the case of pair-end reads.

This provides a way for us to group the variant sites by depth, following which we use the appropriate depth negative log-likelihood table to obtain the negative log-likelihoods for each pair and then finally combine the negative log-likelihoods to get a result for the entire dataset. Additionally, we have introduced a novel weighted sum when calculating the composite negative log-likelihood across coverage depth (Eq 2). Rhometa thus enables the application of the composite likelihood estimator approach for current shotgun metagenomic datasets.

An important advantage of Rhometa and its use of raw reads over a consensus assembly from each sample, is that the potential microdiversity within each dataset is preserved for analysis.

As discussed, Rhometa performs better the more genomes there are, it is possible to get a minimum count for the number of genomes present when simulated under the coalescent model with recombination. In real metagenomic samples, any single sample may have millions of genomes of the same species, and across samples there may be significant population structure that is not captured by the standard coalescent model with recombination. The relationship between the number of metagenomic samples, the depth of sequencing of each sample, and the genome count in our simulation study is therefore not straightforward.

## Limitations and future directions

While we have endeavoured to make a complete package with Rhometa that addresses all aspects of population recombination rate, there are a few limitations. One such limitation is

the automatic inference of the tract length, which is also not possible with LDhat [12] or PIIM [22]. In the context of the composite likelihood approach, the authors of both LDhat and PIIM suggest that while it may be theoretically possible to co-estimate the population recombination rate and tract length, in practice it is challenging. Instead, following the examples of LDhat and PIIM, Rhometa fixes the mean tract length for population recombination rate estimation. As observed by the authors of PIIM, tract length tends to rescale the population recombination rate estimate and large misspecifications can cause further deviations [22]. In our tests with simulated datasets where the tract length was varied, it was found that misspecifying the tract length tends to linearly scale the final estimates in a predictable manner (S6 Fig). If a fixed value of tract length is given by the user for all analyses (i.e. an accurate sample-specific tract length is not provided by the user), then differences in inferred rho among samples could be due to either a change in the recombination rate or a change in tract length. The user will need to take this into account when interpreting the results. For instance, if performing same species comparisons across time and location, the tract length can be fixed to determine relative changes in the magnitude of recombination.

Furthermore, the nature of our method is not sensitive to very low rates of recombination as observed when attempting to evaluate rates between 0–1 and we suggest exercising caution for such fine scale analysis.

Another point of note concerns the generation of the lookup tables for the program. While it is relatively fast to generate lookup tables due to the incorporation of LDpop, it can still require substantial time and resources for a high-resolution table with a large number of genomes. Generation of lookup tables require specification of, in our tests we have found that Rhometa is fairly robust to misspecifications of $\theta$. It is only extreme misspecifications, such as misspecifications by a factor of 10, that seem to have a large impact (S5 Fig).

We believe the availability of a tool such as Rhometa, which can be easily applied to current metagenomic datasets is timely and significantly expands the range of habitats and therefore microbial communities that can be studied for recombination, giving us an insight into the extent to which they can adapt and speciate. How $\rho$ varies within environments and between taxa is unknown, Rhometa can help investigate many such fundamental questions related to the evolution and survival capacity of microbes. With the aid of data analysis techniques, metagenomic datasets can be further combined with environmental and sequencing metadata to help study the intricacies of recombination. Many ecological factors can modulate and effect recombination [48]. Synthesis of other data types with the results of our program may yield a clearer understanding of such relationships. We have built our approach in a modular and easy to adapt manner making this and similar applications easy to explore in the future.

## Supporting information

**S1 Fig. Results of varying simulated genome lengths for testing LDhat (number of genomes fixed at 10, tract length 500).**
(TIF)

**S2 Fig. Comparing simulated single end and paired end read datasets in Rhometa.** (**A**) Single end results. (**B**) Paired end results.
(TIF)

**S3 Fig. Simulated vs Estimated population recombination rate ($\rho$) results for Rhometa.**
Results for varying numbers of simulated genomes and fold coverage values for population recombination rates 0.0, 0.1, 0.2, 0.3, 0.4, 0.5. The simulation parameters used are the same as for population recombination rates [10.0, 20.0, 30.0, 40.0, 50.0], except lookup tables for

population recombination rates 0–2 were used (0–2 in 201 steps) for depths of 3–200.
(TIF)

**S4 Fig. Simulated vs Estimated theta per site (θ) results for LDhat and Rhometa.** (**A**) LDhat. Simulated vs Estimated theta per site (θ) for varying number of simulated bacterial genomes. (**B**) Rhometa. Simulated vs Estimated theta per site (θ) for varying number of simulated bacterial genomes.
(TIF)

**S5 Fig.** Results of analyzing simulated datasets with lookup tables generated under misspecified θ (genome length fixed at 100,000) (**A**) Simulated datasets analyzed with true θ (0.01) lookup tables (**B**) Simulated datasets analyzed with misspecified θ (0.005) lookup tables (**C**) Simulated datasets analyzed with misspecified θ (0.02) lookup tables (**D**) Simulated datasets analyzed with misspecified θ (0.1) lookup tables.
(TIF)

**S6 Fig. Results of analyzing simulated datasets with misspecified tract lengths (genome length fixed at 100,000.** θ 0.01 lookup tables used) (**A**) Simulated datasets analyzed with true tract length 1000 bp (**B**) Simulated datasets analyzed with misspecified tract length 500 bp (**C**) Simulated datasets analyzed with misspecified tract length 2000 bp.
(TIF)

**S7 Fig. Mcorr tested using the simulated datasets used for rhometa (Figs 3 and 4).** Mcorr uses bam (aligned reads) and gff (gene annotations) files as inputs. For the gene annotations, coding regions need to be provided for mcorr. With the simulated data every 1000 bp was defined as a coding region (CDS). Out of 4000 simulations, some could not be completed and many extremely large unrealistic realistic values. A filter was used where such instances were discarded and only phi_pool values < 100 were kept. After filtering there were only 2386/4000 values.
(TIF)

**S1 Table. Analysis results of s_pnemoniae transformation.** Results for all seed values.
(DOCX)

**S2 Table. *S.pneumoniae* experiment accession codes.**
(DOCX)

**S3 Table. Accession codes for Port Hacking, Sydney datasets (from SRA run table).**
(DOCX)

**S1 Appendix. Lookup configuration.** Matching against the lookup table.
(DOCX)

## Acknowledgments

I would like to acknowledge Dr. Michael Lake of UTS eResearch for helping me with HPC related tasks using which all the analysis in this paper was performed. I would also like to thank Fredrick Jaya, PhD Student at ANU for helping with the help message function in the nextflow pipelines.

## Author Contributions

**Conceptualization:** Matthew Z. DeMaere, Aaron E. Darling.

**Data curation:** Sidaswar Krishnan.

**Formal analysis:** Sidaswar Krishnan, Matthew Z. DeMaere.

**Funding acquisition:** Aaron E. Darling.

**Methodology:** Sidaswar Krishnan, Matthew Z. DeMaere, Aaron E. Darling.

**Project administration:** Justin R. Seymour, Aaron E. Darling.

**Resources:** Aaron E. Darling.

**Software:** Sidaswar Krishnan, Matthew Z. DeMaere, Aaron E. Darling.

**Supervision:** Matthew Z. DeMaere, Dominik Beck, Justin R. Seymour, Aaron E. Darling.

**Validation:** Sidaswar Krishnan, Matthew Z. DeMaere, Aaron E. Darling.

**Visualization:** Sidaswar Krishnan, Martin Ostrowski.

**Writing – original draft:** Sidaswar Krishnan.

**Writing – review & editing:** Matthew Z. DeMaere, Dominik Beck, Martin Ostrowski, Justin R. Seymour, Aaron E. Darling.

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
