## [Decision Letter · Decision Letter 0]

18 Oct 2022

Dear Dr DeMaere,

Thank you very much for submitting your Methods entitled 'Rhometa: Population recombination rate estimation from metagenomic read datasets' to PLOS Genetics.

The manuscript was fully evaluated at the editorial level and by independent peer reviewers. The reviewers appreciated the attention to an important problem, but raised some substantial concerns about the current manuscript. Based on the reviews, we will not be able to accept this version of the manuscript, but we would be willing to review a much-revised version. We cannot, of course, promise publication at that time.

If you decide to revise the manuscript for further consideration at PLOS Genetics, please aim to resubmit within the next 60 days, unless it will take extra time to address the concerns of the reviewers, in which case we would appreciate an expected resubmission date by email to plosgenetics@plos.org.

We are sorry that we cannot be more positive about your manuscript at this stage. Please do not hesitate to contact us if you have any concerns or questions.

Yours sincerely,

Xavier Didelot

Academic Editor

PLOS Genetics

Xiaofeng Zhu

Section Editor

PLOS Genetics

Reviewer's Responses to Questions

**Comments to the Authors:**

Reviewer #1: In this article, the authors have introduced a method to infer the population recombination rate in prokaryotic populations, which is novel and can be very useful in a bunch of analysis. The authors built up a pipeline of going from metagenomic data of reads to vcf files to plug into the framework of LDpop, with a gene-conversion based recombination model. In brief, I think the method and analysis in this article is really solid and the software can be really promising in a lot of studies. Here are some of my minor comments to this manuscript:

1. In line 123 you said that Pyrho is a method supporting read-based data but is only workable for diploid genomes. However, as far as I understand, the basic methodology of Pyrho is not that different from LDhat or LDpop except that Pyrho can handle a population size history and I believe the input format needs to be a vcf file instead of reads data directly. I recommend correcting the mention of Pyrho here (if what I recall is correct), and really mention the contribution of your work on handling the bioinformatics of read data and the modules in the pipeline.

2. I might have missed this but you mentioned using the approx option while generating the lookup table from LDpop, did you benchmark the performance of inference when generating the table using approx, especially for larger sample size? It would be nice to see some simulation-based benchmarking results while having a large sample size and using approx option to give people a sense of how much accuracy is lost by doing this.

3. Did you investigate sequencing errors while simulating reads? Sequencing errors can give false signals to recombinations and thus confound the analysis. I would suggest investigating the robustness of the method under different error rates (close to real world sequencing technology parameters).

Overall I think this work is really solid and the authors have made good points in the use of the method. I hope this could contribute to the field and facilitate a lot of studies!

Reviewer #2: Review of "Rhometa: Population recombination rate estimation from metagenomic read datasets"

This paper introduces a new computational pipeline for estimating recombination rates in metagenomic datasets. The overall goal is very important - as the authors emphasise, the majority of microorganisms cannot be cultured and hence we only observe them in in their natural, complex, environments via metagenomics. The software pipeline has been well explained and care has been taken to ensure that it is appropriate and can run at scale.

Unfortunately I have concerns with the paper as it stands:

Firstly, the writing is far from the expected standards. I've described many specifics below but overall it needs a complete and thorough overhaul from an experienced scientific writer. In places this gets in the way of review as I could not quite follow the details. Except in the limitations section (which seemed strong), there was little in the way of reflection or consideration about the details, and the text needs to provide more insight and less technical repetition.

To give an example from the abstract: "the deviation from the expected value decreased as the number of genomes increased" is about all that is claimed, but this is just one minor pre-requisite for all publishable methods... What can we say that is meaningful about its performance? How does it compare to the state of the art? Does it do something that is currently impossible otherwise? The same writing problem exists throughout.

On Line 621 it is explained that the essence of this paper is that "for aligned read based datasets the main complication is that the depth can vary greatly from site to site." This is conceptually very straight forward, and just requires (a lot of!) careful technical work to implement (for great value, I agree). That intuition should come through more cleanly thoughout (starting in the abstract).

Secondly, I did not find the analysis of the data to be completely compelling. Results are only given for "rho", the population recombination rate, in one simulation scenario (varying true rho and data quantity). I would have expected some sensitivity analysis around critical parameters - the average tract length, but also population demography, variation of rho along the genome, presence of related strains, etc.

I'd like head to head comparisons where this is possible. However, since this is arguably the first method to do what it does, it doesn't matter if some of the experiments fail - but this information is critical to users for interpretation. (the slight bias in the inference for high rho is a case in point.)

I did not find the real data analysis convincing to the central claim that the method was better than its rivals, but that may be because I could not follow the exposition around it. Details are below.

This is not a criticism, but I am unsure what to say about the novelty of this work. What novelty there is here is in putting together work into a well functioning pipeline to solve a particular scientific need. I would like to support the paper against such a criticism because I do think there is huge value in making these tools well from a software engineering perspective, and having venues for them to be published.

Finally I'd like to see something about computational requirements, for users to get a scale for when this is practical.

Details and minor points:

Abstract "In simulated datasets, the deviation from the expected value decreased

as the number of genomes increased " is hardly a headline result. I suggest replacing with something more specific.

Summary /Abstract: say something about the scale of data your method is suitable for.

Line 87,89: I found the final units unclear “rate of initiation of gene conversion” and “mutation rate”. Indeed the line below confuses the matter as we then divide by the mutation rate.

I did not find the separation of the Mcorr method convincing. The description is confuised and doesn't make it clear exactly what Mcorr is estimating and why we would prefer what Rhometa does.

Line 153: "We have carefully adapted the program to work with modern read based metagenomic datasets." Elsewhere it is implied that this is new software, and I think it probably is? Be very clear about where the code is coming from. (It is natural that the ideas behind LDhat would be incorporated, and this is and should be acknowledged.)

Wattersons Theta estimate will be biased after accounting for alignment and recombination, compared to the estimate of theta that would be obtained by a likelihood-based approach (which could be the "true parameter value" from some model). Typically this would be handled by an iterative algorithm and usually does not take many iterations? At least the bias needs discussing.

The section "Lookup table and depth" is very fuzzy. What specifically do you do? It sounds like there is a threshold (minimum depth?) to which you subsample down to? Or is this entirely optional?

Equation 1 is unclear. I think your \\underline{t} notation is misleading? (Its described as a mean distance so I think so?) Underline would normally indicate a vector, overline for mean but since it is a parameter I don't think that is appropriate either.

Line 261 "we use the matched likelihood values for recombination rates" what does this specifically mean? Generally this paragraph is hard to parse.

Line 269 "we introduce a novel weighting algorithm" you haven't motivated this specific form for the weighting. Does it approximate some sensible procedure, like a likelihood interpretation, or the standard deviation of the estimate for the corresponding normal normal distribution approximation? Does it appear elsewhere? Why this form?

"maximum log likelihood value (closest to 0) corresponds to the most likely final estimated population recombination rate" Firstly, can we get it clear whether this is a likelihood or not (perhaps a negative log-likelihood?) Secondly, "most likely final estimated" is very unclear.

(Line 278 what happened to line numbers here? )

Minor typography note: Throughout we've labelled "Equation X" but referred to them in text as "equation"

Figure 1: I'm not quite sure how to map the content to the text. For example, "Estimate theta (Python)" does not tell me whether this is the Wattersons Theta estimate. I think it is the (new?) Python components that need clarification, but you might introduce labels for all components. (I found that in the description L309, but the criticism still stands in trms of clarity.)

Line 362: "population_size was 1" what scaling is this? Does it mean something?

Line 366: "The recombination events were simulated first then mutation events were simulated on top" - I assume this is referring to the method that msprime uses for neutral mutations. Either explain properly, or omit as it is standard.

Line 367 Notation: "calculated as such"

Line 373 very verbose and not particularly clear or precise.

Line 384: Can we have a simple version here, and relegate the details to the methods?

Para at Line 364: Typeset the equations.

The text contains the text "rho" and "theta" instead of the symbols throughout.

L437 "385 is the number that is thought to be false positives" how was this determined? If this number comes from elsewhere, just give the equation as Expected true variants/Total genome length.

Another way of putting the simulated Rhometa results is that reegardless of the number of genomes, \\rho is biased. This can be ok, but be up front with it.

Line 506-514 Don't just tell us the numbers in the table. Either leave them in the table, or interpret them.

Line 518-525: Do you really mean that your estimate differs from mcorr by a factor of 2000? Or is there some scaling going on? I don't understand wher the r/m=380 estimate in the next paragraph comes from, is this not their estimate?

Line 527 You've given a very long paragraph containing many estimates and I don't understand the sources of the data for any of them properly.

Line 606(and much of this paragraph) is far too technical for a discussion. The purpose of this paragraph does not come through; its just more results without context.

Reviewer #3: The paper is very well written and contains some interesting ideas for estimating recombination rates in microbial populations, however, I have a few concerns regardging the approach and validation.

1. The authors have claimed that: "The best way to study recombination in a microbial population is via metagenomics, which allows us to study microbes in their natural environment via direct sequencing and analysis of environmental DNA [7,8]"

The phrasing of this sentence is misleading to a reader because both [7,8] do not discuss microbial recombination. This also leads me to question the general claim: 'microbial recombination is best studied using metagenomics'; is it (currently) well supported by the literature?

Additionally, I am (personally) a bit skeptical about this idea in general. Apart from the obvious complexities associated with shotgun metagenomic data such as the mixing of data from multiple sources, based on the literature, metagenome assembled genomes (MAGs) and metagenomic data in general can also suffer from serious data quality issues (for example, see https://doi.org/10.1093/bib/bbac413 and https://doi.org/10.1093/gbe/evaa238). While overcoming such data quality issues is not the scope of this paper, how would rhometa behave in the face of these concerns? How reliable will the recombination estimates be? In particular, given the fact that a metagenomic dataset will most likely contain a mixture of data from sources undergoing different evolutionary processes governing recombination, what exactly would a point estimate for recombination represent? Wouldn't a method that uses processed, carefully curated and filtered data be better suited for estimating microbial recombination?

The authors should clearly address this point.

2. Although rhometa is designed to be used with metagenomic data, the S. pneumo dataset used for its validation is not a metagenomic dataset and the reference ATCC700669 (NC_011900.1) is not a MAG. The authors must have used this dataset for direct comparison with mcorr. However, I feel that rhometa should be validated using real metagenomic data. I will be very interested to see how rhometa behaves when faced with real-world complexities and potential data quality issues that I mentioned in my previous comment.

Since there is limited availability of such experimentally estimated recombination rates for published metagenomic datasets, I think the authors can instead follow a cross-validation process to benchmark rhometa. Although we will have no means to judge the prediction accuracy, assuming rhometa works well in the presence of aforementioned challenges, the repeated estimates should have high precision.

3. Additionally, the simulated data used to validate RhoMeta are also completely 'noise-free' and not technically 'metagenomic'. In fact, the entire simulated data validation pipeline seems far too simplistic for a tool designed for metagenomic data. Additionally, I am not sure if msprime is the best tool to simulate data for this purpose at all, for example, see section 2.2. of https://doi.org/10.1093/bioinformatics/bty630 for a list of tools that might be better suited for this purpose. In addition, the authors should also properly benchmark and report performance metrics (CPU, RAM requirements, etc.), including its limitations in terms of maximum genome size, maximum number of sequences, etc. (as applicable).

4. Finally, the inability to infer the recombination tract length could be a potentially major pitfall. Firstly, the recombination estimate seems to depend on this value, and secondly, since rhometa is intended to be used with metagenomic data, a potential user will find it nearly impossible to rely on the previous literature to find a 'guess' for this parameter. Therefore, the authors should clearly outline a process to estimate or guess this value.

**Have all data underlying the figures and results presented in the manuscript been provided?**

Reviewer #1: None

Reviewer #2: Yes

Reviewer #3: Yes

PLOS authors have the option to publish the peer review history of their article (what does this mean?). If published, this will include your full peer review and any attached files.

Reviewer #1: **Yes: **Yun Deng

Reviewer #2: No

Reviewer #3: No

---

## [Editor Report · Decision Letter 1]

27 Feb 2023

Dear Dr DeMaere,

We are pleased to inform you that your manuscript entitled "Rhometa: Population recombination rate estimation from metagenomic read datasets" has been editorially accepted for publication in PLOS Genetics. Congratulations!

Yours sincerely,

Xavier Didelot

Academic Editor

PLOS Genetics

Xiaofeng Zhu

Section Editor

PLOS Genetics

Comments from the reviewers (if applicable):

**Data Deposition**

http://datadryad.org/submit?journalID=pgenetics&manu=PGENETICS-D-22-00986R1

**Press Queries**

---

## [Editor Report · Acceptance letter]

20 Mar 2023

PGENETICS-D-22-00986R1 

Rhometa: Population recombination rate estimation from metagenomic read datasets 

Dear Dr DeMaere, 

We are pleased to inform you that your manuscript entitled "Rhometa: Population recombination rate estimation from metagenomic read datasets" has been formally accepted for publication in PLOS Genetics! Your manuscript is now with our production department and you will be notified of the publication date in due course.

With kind regards,

Anita Estes

PLOS Genetics

On behalf of:
